# Oncolytic Herpes Simplex Virus-Based Therapies for Cancer

**DOI:** 10.3390/cells10061541

**Published:** 2021-06-18

**Authors:** Norah Aldrak, Sarah Alsaab, Aliyah Algethami, Deepak Bhere, Hiroaki Wakimoto, Khalid Shah, Mohammad N. Alomary, Nada Zaidan

**Affiliations:** 1Center of Excellence for Biomedicine, Joint Centers of Excellence Program, King Abdulaziz City for Science and Technology, P.O. Box 6086, Riyadh 11451, Saudi Arabia; Naldrak@kacst.edu.sa (N.A.); smalsaab@kacst.edu.sa (S.A.); asalgethami@kacst.edu.sa (A.A.); 2National Center for Biotechnology, Life Science and Environmental Research Institute, King Abdulaziz City for Science and Technology, P.O. Box 6086, Riyadh 11451, Saudi Arabia; 3Center for Stem Cell Therapeutics and Imaging (CSTI), Brigham and Women’s Hospital, Harvard Medical School, Boston, MA 02115, USA; DBHERE@BWH.HARVARD.EDU (D.B.); HWAKIMOTO@mgh.harvard.edu (H.W.); KSHAH@bwh.harvard.edu (K.S.); 4Department of Neurosurgery, Brigham and Women’s Hospital, Harvard Medical School, Boston, MA 02115, USA; 5BWH Center of Excellence for Biomedicine, Brigham and Women’s Hospital, Harvard Medical School, Boston, MA 02115, USA; 6Department of Neurosurgery, Massachusetts General Hospital, Harvard Medical School, Boston, MA 02114, USA; 7Harvard Stem Cell Institute, Harvard University, Cambridge, MA 02138, USA

**Keywords:** oncolytic herpes simplex virus (oHSV), virotherapy, immunotherapy, cancer

## Abstract

With the increased worldwide burden of cancer, including aggressive and resistant cancers, oncolytic virotherapy has emerged as a viable therapeutic option. Oncolytic herpes simplex virus (oHSV) can be genetically engineered to target cancer cells while sparing normal cells. This leads to the direct killing of cancer cells and the activation of the host immunity to recognize and attack the tumor. Different variants of oHSV have been developed to optimize its antitumor effects. In this review, we discuss the development of oHSV, its antitumor mechanism of action and the clinical trials that have employed oHSV variants to treat different types of tumor.

## 1. Introduction

Cancer remains a serious global burden that threatens millions of lives worldwide. It is considered the second most common cause of death, with an annual worldwide incidence of 10 million cases [1]. The most common cancer therapies include surgery, chemotherapy, and radiotherapy [2]. Such traditional therapeutic approaches have detrimental consequences such as hematological toxicity, gastrointestinal toxicity, ototoxicity, hepatotoxicity, cardiotoxicity, neurotoxicity, and endocrine fluctuations [3]. Therefore, searching for alternative approaches that specifically target cancerous cells and are safer with no or moderate side effects is a priority for researchers around the globe. In particular, the use of engineered oncolytic viruses (OVs) is a promising alternative approach to specific targeting of tumors due to their ability to efficiently replicate in tumor cells [4]. Additionally, OVs promote a state of anti-tumor immunity [5]. Several OVs have been tested in clinical trials, such as herpesviruses, retroviruses, adenoviruses, vaccinia virus, and poliovirus [6]. These viruses vary in their sizes, genome nature, and replication efficiency [7].

Oncolytic herpes simplex viruses (oHSV) are among the few oncolytic viruses that were moved to phase III clinical trials. Herpes simplex virus (HSV), which belongs to the Herpesviridae family, has a large dsDNA (152 kb) which is encapsulated within an enveloped icosahedral capsid. HSV is a neurotropic virus with two variants: HSV-1 and HSV-2. They possess two significant features, which make them suitable candidates for targeting tumors. Firstly, the genome has approximately 30 kb encoding for nonessential genes [8]. This feature allows for genetic manipulation by adding or replacing genes through genetic recombination. Secondly, herpesviruses have a good safety profile, since they replicate in the nucleus without causing insertional mutagenesis [6]. According to their replication strategies, genetically engineered HSV-1 vectors are categorized into three groups: (i) replication defective, (ii) conditionally replicating and (iii) amplicons. Of these, conditionally replicating HSV-1 vectors are designed to specifically target cancerous cells. Moreover, they are engineered to have therapeutic transgenes to enhance anticancer immunity [9]. Several genetically modified variants of oHSV-1, such as NV1020 [10], G207 [11], T-VEC [12] and HSV1716 (Seprehvir) [13], have been tested extensively in clinical studies. Among these, FDA-approved T-VEC (trade name: Imlygic™) is the most widely used OV. As of December 2019, the clinical trials portal (https://clinicaltrials.gov/) has listed more than 38 clinical trials assessing the safety and effectiveness of T-VEC against several types of cancers. T-VEC was genetically engineered to delete the neurovirulence genes and the genes required for inhibition of antigen presentation [14,15]. Kaufman et al. reported that T-VEC was associated with high levels of melanoma-specific CD8^+^ T cells and decreased levels of immunosuppressive cells such as CD4+ FoxP3+ regulatory T cells [16]. Although T-VEC showed an excellent safety profile, common side effects include fever, nausea, vomiting and headache [17]. To provoke the antitumor immunity, two copies of granulocyte-macrophage colony-stimulating factor (GM-CSF) gene were engineered to replace the deleted neurovirulence gene [18]. Hence, in the current review, the following topics will be covered: (i) the major mechanisms by which oHSV induces tumor regression, (ii) the ongoing preclinical and clinical trials of oHSV, (iii) the role of oHSV in immunovirotherapy, and (iv) the limitations and ethical concerns regarding oHSV.

## 2. Mechanism of Oncolytic Herpes Simplex Virus-1 Antitumor Activity

### 2.1. Tumor-Selective Replication

HSV-1 is a double-stranded DNA virus that has a large genome containing 152 kb, 30 kb of which are dispensable for viral infection [19]. This makes HSV-1 an attractive candidate for genetic manipulation to enhance tumor-selectivity and patient safety. Although its site of replication is the nucleus, HSV-1 does not cause insertional mutagenesis and is sensitive to antiviral drugs such as acyclovir and ganciclovir [20]. Some of the most frequently modified non-essential viral genes in HSV-1 variants that have shown promise in preclinical and clinical studies include γ34.5, UL39, and α47, encoding infected-cell protein (ICP) 34.5, ICP6 and ICP47, respectively. These genes have evolved to provide wild type (WT) HSV-1 with the abilities to evade the host antiviral response and continue its replication cycle. In this section, we will discuss the function of these genes, and how manipulating them can play a role in conferring tumor selectivity.

γ34.5 is one of the most commonly mutated genes for creating oHSVs. It encodes ICP34.5, which allows WT HSV-1 to overcome the host cell protein synthesis shut-off response to viral infection. In WT HSV-1 infection, a normal cell activates protein kinase R (PKR) which inactivates eukaryotic initiation factor-2α (eIF-2α), leading to a shut down in protein synthesis. However, the presence of the ICP34.5 protein allows WT HSV-1 to restore protein synthesis by restoring eIF-2α function [21]. Therefore, deleting both copies of ICP34.5 hinders the HSV-1 ability to synthesize protein and subsequent propagation in normal cells. In contrast, the activation of PKR in tumor cells is often impaired, providing a preferential environment in which γ34.5-lacking HSV-1 can continue its life cycle. To improve safety and reduce the chance of reversion to WT, deletion of γ34.5 can be accompanied with UL39 inactivation (see below) to create G207, the first oHSV to be tested in a clinical trial in the US [19]. ICP6 is the large subunit in viral ribonuclease reductase, which converts ribonucleotides into deoxyribonucleotides that are utilized in viral genome synthesis [22]. Lacking ICP6 restricts viral replication to dividing cells, as mature, postmitotic cells lack ribonucleotide reductase expression and sufficient amounts of deoxyribonucleotides [19].

As an immune-evasion mechanism, HSV-1 infection leads to a down-regulation of MHC class I expression via binding of ICP47 to the transporter associated with antigen presentation (TAP), blocking the antigenic peptide transport in the endoplasmic reticulum and subsequent loading onto MHC class I molecules and presentation on the cell surface [23]. This attenuates the overall immune response to infection through attenuating CD8^+^ T cell recognition of infected cells [24]. Designing a multimutated oHSV and deleting the α47 gene prevents the down-regulation of MHC class I which would allow CD8^+^ T cells to recognize infected tumor cells and hinder viral immune-evasion. G47Δ is a multimutated oHSV variant built from G207, in which, in addition to the deletion of both copies of the γ34.5 gene, an inactivation of ICP6, it contains a deletion of the α47 gene [25]. G47Δ has shown significantly higher efficacy in vivo compared to its precursor G207 at inhibiting tumor growth in immune-competent and immune-deficient animal models [25].

### 2.2. Activation of Innate Immune Responses

Exploiting the immune system to fight cancer by activating the innate and adaptive immune pathways has been studied widely over the last years. Immunotherapy is found to be a promising approach for many forms of cancer in preclinical and clinical trials [19]. Such immunotherapies include chimeric antigen receptor T cell (CAR-T cell), bispecific killer cell engager (BiKE) [26] and oncolytic virotherapy [27]. Their most important mechanism is to activate immune cells such as T cells, natural killer (NK) cells and dendritic cells (DC). Oncolytic virus’ ability to selectively replicate inside tumor cells and stimulate innate immune cells could override the suppressive tumor microenvironment by inducing both antiviral and antitumor activity [20].

Oncolytic herpes simplex virus has been identified to induce a dual mechanism inside tumor cells. These mechanisms, as shown in Figure 1, involve the activation of antiviral pathways by triggering cell death signaling cascades and the induction of host antitumor immune responses by recruiting and activating the surrounding immune cells, which ultimately leads to tumor killing [20].

Recent research has shown that oncolytic viruses, particularly herpes simplex virus, induce cell death that elicits an immune response, identified as immunogenic cell death (ICD) [28]. The level of immunogenicity is associated with the activation of danger signaling pathways and the expression of damage-associated molecular patterns (DAMPs) in the tumor microenvironment [29]. One study has shown that the infection of oncolytic HSV-1 within squamous cell carcinoma (SCC) cells, in vitro, increased the release of ATP and high mobility group box 1 (HMGB1) compared to uninfected SCC. In addition, it induced the translocation of calreticulin (CRT) to the cell surface to act as an “eat me” signal [29].

During viral infection by oHSV, dying tumor cells consequentially release molecular components including ATP and HMGB1, in addition to the surface exposure CRT to prepare the cell for phagocytosis. The recognition of DAMPs by immature dendritic cells (DCs) facilitates their maturation and infiltration. Dendritic cells mature as they infiltrate into the tumor microenvironment in response to inflammatory chemokines such as CC-chemokine ligand 4 (CLL4) [28]. While DAMPs can prompt DCs maturation, the chemokine receptor expression and chemokine responsiveness can play an important part in DCs’ activation, cross-presentation, and migration to the draining lymph nodes [27]. The stimulation and recruitment of local macrophages and dendritic cells triggers them to engulf dying tumor cells and to process internalized tumor antigens to be presented to naive T cells. This results in priming the cytotoxic effect by T cells, and therefore leads to an antitumor response [28]. Additionally, the recognition of DAMPs and viral components by the innate sensing via pattern recognition receptors (PRRs) stimulates type I interferon production via stimulator interferon genes (STING) signaling pathway. In a recent study, it was found that the STING-dependent pathway has a critical role in the spontaneous priming of antitumor T cells [30]. This provides insights into cancer immunotherapy indications, which can be used for the development of current cancer immunotherapies.

### 2.3. Activation of Adaptive Immune Responses

The activation of T cells is a crucial part of a cell-mediated adaptive immune response. Sufficient activation of innate immunity and the cross-presentation, which is the process of presenting foreign antigens on MHC class I molecules, are essential for the activation of CD8^+^ T cells [20,31]. Activated CD8^+^ T cells clonally expand in secondary lymphoid organs such as the spleen and lymph nodes, and then, aided by chemokine gradients, migrate to the sites of inflammation [27,28,32]. As the tumor microenvironment (TME) suppresses CD8^+^ T cell trafficking and function due to multiple effects, oncolytic virotherapy offers the potential to overcome this by eliciting an adaptive immune response both against the viral infection and the tumor [33].

Tumors have evolved multiple immune-evasive mechanisms to interfere with the stimulation of a tumor-specific T cell response [34]. In oncolytic virotherapy, the initial T cell response may be instigated by oncolytic viral activity; however, the presence of tumor-associated antigens and tumor cell debris in the inflammatory milieu of the TME promotes the cross-presentation of tumor antigens to CD8^+^ T cells [28]. As certain oncolytic herpes simplex viruses are engineered to have a deletion in both copies of ICP47, which is a protein that interferes with the normal processing and presentation pathways of MHC class I, levels of MHC class I in infected cells are enhanced [19,35], which prevents viral inhibition of tumor-associated antigen processing and presentation [28]. In a study by Benencia et al., 2008, it was shown that ICP47-lacking oHSV significantly increases tumor antigen uptake by DCs, leading to an enhanced effect of tumor-associated NK and CD8^+^ T cells [36].

Oncolytic viruses can be engineered to express genes that enhance the immune response. Most commonly, oncolytic viruses may be armed with cytokines, which are immunomodulatory genes that play an essential role in the recruitment and homeostasis of T cells [28]. Cytokine can also sustain CD8^+^ T cell activation, contributing to an enhanced local and distant antitumor response [37]. In a phase II clinical trial using OncovexGM-CSF for melanoma, 26% of the patients with unresectable stage III or IV melanoma demonstrated objective clinical response, which included regression of infected and distant noninfected lesions [16].

As opposed to innate immunity, adaptive immunity induces immune memory, meaning that recurrent exposure to the same antigen will generate a stronger response [20]. Fully functional CD8^+^ T cells maintain ongoing tumor-specific surveillance against distant tumors or potential relapse [34]. As a result, when using oncolytic virotherapy, it is important to take antiviral memory into consideration, as it may hinder retreatment [20]. Oncolytic viruses may naturally induce an antiviral immune response, which may lead to clearance of the viruses prior to them performing their intended function [6]. Therefore, the balance between viral immunogenicity and antitumor immunity is crucial in determining the efficacy of oncolytic virotherapy [28].

## 3. oHSV Derivatives in Pre-Clinical Models

Various genetically engineered oHSVs have been developed for the treatment of different cancers [38]. Multiple approaches have been employed to enhance the effects of oHSVs [28]. Such approaches include arming oHSV with pro-drug metabolizing transgenes such as thymidine kinase [39], proapoptotic genes such as TNF-related apoptosis-inducing ligand (TRAIL) [40,41], or immunostimulatory transgenes such as GM-CSF [14,16]. Indeed, various preclinical studies have demonstrated the therapeutic efficacy of different oHSV variants in treating multiple types of tumors [38]. Table 1 summarizes several oHSV variants, their genetic modification and the preclinical models they have been tested on.

rRp450, an oHSV variant with an ICP6 deletion and an insertion of rat cytochrome P450 2B1 (CYP2B1), a pro-drug enzyme for cyclophosphamide (CPA), has initially shown therapeutic efficacy in treating preclinical animal models of glioblastoma [42]. In 2002, Pawlik et al. studied the efficacy of rRp450 in treating diffuse colon carcinoma liver metastases. The study showed that treatment with rRp450 and CPA significantly decreases the tumor burden from uncountable metastatic nodules in the control group to five in the treated group [43]. In an aggressive sarcoma mouse model, the administration of rRp450 and CPA significantly increased mouse survival compared to the control group [44]. In an orthotopic mouse model of atypical teratoid/rhabdoid tumors, treatment with rRp450 and CPA significantly prolonged the median survival to 84.5 days in tumor-bearing mice compared to 50 days in the vehicle-treated controls [57].

Another oHSV variant is rQNestin34.5v.2, which is an oHSV that retains ICP34.5 expression under the nestin promoter/enhancer elements [58]. This variant has demonstrated preclinical therapeutic efficacy for neurological tumors. Upon treatment with rQNestin34.5v.2, 77.8% of athymic mice bearing intracerebral human U87dEGFR glioma tumors survived >90 days, whereas mice injected with vehicle only survived up to day 21 post-tumor implantation [45].

Initial studies involving the second generation oHSV variant G207, which is genetically modified with a diploid deletion in γ34.5 and an inactivation in UL39, showed efficacious treatment of gliomas [46]. Combining G207 with fractionated ionizing radiation resulted in a synergistic action, as seen with reduced tumor sizes and prolonged survival in mice bearing high-grade gliomas [47]. In a xenogeneic flank mouse model of cervical cancer, a single injection of G207 led to the reduction in tumor burden by 50% [48].

Preclinical studies using G47Δ, which is a third generation oHSV carrying a γ34.5 diploid deletion, UL39 inactivation and an α47 deletion, demonstrated its efficacy and safety in different tumor models including brain [49], gastric [50], liver [51], breast [52], thyroid [51], and urological [52] cancers. In a study conducted by Nigim et al., mice bearing patient-derived malignant meningioma were treated with two injections of G47Δ. This treatment significantly prolonged survival, with 20% of mice surviving >160 days. Furthermore, the authors reported a lack of signs of encephalitic associated with G47Δ treatment, which confirms the safety of this treatment [49]. Several studies have tested the efficacy of G47Δ in treating types of breast cancer [52]. A study by J. Wang et al. reported a 9-fold reduction in the number of metastatic breast cancer nodules in the lungs of G47Δ-treated group compared to the control-treated group [59].

oHSV can be engineered to express proapoptotic or immunostimulatory genes to enhance the antitumor effect [28]. A G47Δ oHSV variant that expresses TRAIL was shown to significantly inhibit glioblastoma multiform tumor growth and invasion, and prolonged mice survival [41]. oHSV expressing immunostimulatory genes such as IL-12 or GM-CSF have been shown to have therapeutic efficacy via stimulating an antitumor immune response. For instance, IL12-expressing oHSV was shown to promote tumor-specific CD8^+^ T cell responses in the peritoneal cavity and omentum in ovarian cancer models [55]. Ghouse et al. reported that treatment of mice bearing triple-negative breast cancer with oHSV-IL12 resulted in the induction of local and abscopal immune effects, in addition to significantly prolonging the survival of treated mice compared to the control group [56]. GM-CSF-expressing oHSV have demonstrated tremendous potential in treating advanced melanoma, which was later approved by the FDA to treat stage III unresectable advanced melanoma [18].

## 4. oHSV Route of Delivery

### 4.1. Systemic Versus Local Delivery

Whether it is intratumoral or systemic administration, certain hurdles pose a challenge to oHSV delivery [60]. Although intratumoral injection delivers virus particles directly to the tumor, there are multiple limitations that prevent optimal delivery and spread within the tumor microenvironment. For instance, certain tumors may be comprised of several nodules located over a large area, or in an anatomical position inaccessible or inconvenient for local injection [61]. For example, the rates of response in phase I and Ib clinical trials using local injection of oHSV for glioblastoma have been suboptimal, potentially due to the influx of blood and cerebrospinal fluid into the cavity after surgical intervention and tumor resection, that resulting in washing out the virus [40]. Intravenous administration allows the virus to reach both primary and metastatic tumors simultaneously; however, rapid clearance of the viruses from the circulation before they reach their targets by antibodies, antiviral cytokines, and immune blood cells, in addition to nonspecific tissue, are among the main hurdles for systemic administration [62]. To overcome some of these challenges, researchers have explored the use of cell carriers to “cloak” the virus particles from the host’s defense. Some of the tested oHSV cell carriers include neural precursor cells [63], lymphocytes [64] and mesenchymal stem cells [65]. In the next section, we will explore the use of such carrier cells to deliver oHSV.

### 4.2. Mesenchymal Stem Cells as OV Carrier Cells

Mesenchymal stem cells (MSCs) have been explored as potential vehicles for gene therapy in tumors as they have demonstrated preferential integration into sites of tumor development [66,67]. The source of MSCs can be autologous, from patient’s bone marrow or adipose tissues, or allogeneic, from placenta and umbilical cord, which can be used as “off-the-shelf” cells [68]. Preclinical and clinical data indicate the safety of using MSCs, as there were no major health concerns reported, suggesting the relative safety of MSC therapies [69]. Another feature of MSCs is their low immunogenicity due to the lack of co-stimulatory molecules expression, eliminating the need for immunosuppression during allogenic transplantation [70]. All of these features make MSCs an excellent candidate to be used for oncolytic virus delivery to tumor sites.

MSCs have been extensively explored for their ability to “cloak” oncolytic viruses, providing them with effective protection from host neutralizing immunity and effective delivery to the tumor microenvironment [71]. Among the studied oncolytic viruses in conjunction with MSCs are Newcastle disease viruses [68], adenoviruses [72], measles viruses [73], and herpes simplex viruses [74,75]. oHSV has been frequently studied in conjunction with MSCs, and has shown promising results in treating gliomas, metastatic melanomas, breast and ovarian cancers, whether delivered systemically [74,75] or locally [40].

## 5. Updates on Recent Preclinical and Clinical oHSV Immunotherapies

At the beginning of the 1990s, genetically engineered oncolytic viruses were first reported in a pre-clinical mouse model showing selective antitumor effects [76]. Since then, different oHSV mutants have been established, including Oncovex GM-CSF (T-VEC), G207, HSV1716, HF10, NV1020 and G47Δ, to increase its safety and efficacy [11,17,77,78]. Many oHSV variants have entered either phase I, II, or III clinical trials or have completed them to treat different types and degrees of tumor, such as breast cancer, glioma and melanoma [79].

### 5.1. Talimogene Laherparepvec (T-VEC)

T-VEC is the first oncolytic virus to be FDA-approved for unresectable stage III advanced melanoma. It is a modified type 1 herpes simplex virus JS1 strain with ICP 34.5 genes deletion, reducing the neurovirulence of the virus and elevating the selectivity of infection for cancer cell killing [15,18,80]. Additionally, the ICP47 gene is deleted, allowing antigen presentation which increases the expression of US11, causing an increase in ICP 34.5-deleted HSV-1 replication without loss of its tumor specificity [81]. The addition of a human granulocyte–macrophage colony-stimulating factor (GM-CSF) permits local expression, increasing the activation of antigen-presenting cells, enhancing the antitumor immune response [15]. The precise mechanism is still unknown [82].

T-VEC is the only variant that has entered a phase III clinical trial, where it is administered in intralesional sites in advanced melanoma patients, causing a reduction in tumor growth and a systemic anti-tumor effect, prolonging survival rate [83]. Biopsies obtained from patients who received T-VEC showed a lower level of immune-suppressive cells such as CD4^+^ FoxP3^+^ regulatory T cells, CD8^+^ FoxP3^+^ T cells and myeloid-derived suppressor cells (MDSCs) in the TME compared to the untreated controls [84].

### 5.2. Clinical Trials of oHSV

oHSV has been studied extensively and tested in clinical setting with approximately 86 clinical trials to study and test for future clinical translation. Those clinical trials were registered at ClinicalTrials.gov and were accessed on February 13th, 2021. T-VEC is one of the most extensively investigated oHSV variants in clinical settings, as shown in Figure 2.

T-VEC is mostly studied as monotherapy or in combination with immune check inhibitors, radiotherapy, or chemotherapy to treat different types of cancer, although several investigate its effect on melanoma, due to the ease of lesion accessibility [85]. In a phase Ib trial of T-VEC in combination with immune check inhibitor ipilimumab (anti-CTLA4 antibody) for treating unresectable stage IIIB-IV melanoma, combination treatment showed higher efficacy compared to monotherapy, with a 50% objective response rate and no dose-limiting toxicities [86]. In this trial, durable response for over 6 months was seen in 44% of the patients. In a phase I trial assessing T-VEC in combination with pembrolizumab (anti-PD1 monoclonal antibody) for treating melanoma, 62% of the patients showed an objective response and a third exhibited an extraordinary complete response. A randomized phase III trial of this combination is currently under investigation [28]. Other oHSV variants such as G207, C134, and rQNestin34.5 are currently under clinical investigation for their safety and efficacy to treat high-grade glioma [87,88]. Table 2 summarizes most of the clinical trials employing such variants and their clinical studies.

RP1 is an oHSV variant modified to contain two deletions of γ34.5 and ICP47, and expression of GM-CSF and GALV-GP-R-, a fusogenic glycoprotein membrane isolated from gibbon ape leukemia virus [94,95]. Expression of GALV-GP-R- was shown to have tumor cell killing ability and an immunogenic response [95]. This variant is currently under evaluation as monotherapy for treating squamous cell carcinoma (NCT04349436), and in combination with PD-1 inhibitors such as nivolumab (NCT03767348) or cemiplimab (NCT04050436) [94].

ONCR-177 is an oHSV variant that contains mutations in the UL37 and ICP47 genes, which prevents replication and neuropathic activity in normal cells [96]. It is armed with five immune-modulatory agents, FLT3LG, IL-12, CCL4, anti-CLTA-4 and anti-PD-1, to increase T cell and NK activation in addition facilitating CD8^+^ T and dendritic cell recruitment [94,96]. ONCR-177 is in clinical trials to assess its safety and the preliminary anti-tumor efficacy as a monotherapy or in combination with pembrolizumab (anti-PD-1) for treating metastatic solid tumors (NCT04348916).

While most clinical trials investigate variants of oHSV-1, some clinical trials are investigating the safety and therapeutic efficacy of oncolytic variants of herpes simplex virus type-2 (HSV-2). OH2 is a potent oncolytic variant of HSV-2 with ICP34.5 and ICP47 deletion and human GM-CSF expression [79]. Certain variants of OH2 may include a deletion in the protein kinase (PK) domain of the ICP10 gene. The PK domain activates the Ras/MEK/MAPK mitogenic pathways, which facilitates HSV-2 replication. Deletion of this domain prevents HSV-2 replication in normal cells and restricts it to cells with aberrantly activated Ras pathways, such as tumor cells [97]. OH2 expressing GM-CSF has demonstrated significant therapeutic efficacy against metastatic ovarian cancer in preclinical models [98]. OH2 is currently undergoing clinical trials for treating different solid tumors in humans as a monotherapy (NCT04637698) or in combination with other drugs such as PD-1 inhibitor pembrolizumab (NCT04386967) or HX008 (NCT03866525 and NCT04616443).

## 6. Limitations

The route of delivery and the presence of pre-existing immunity are two important factors that may affect the efficiency of oHSV therapeutics. Delivery of oHSV is usually achieved by direct intertumoral injections or by locoregional means. However, these delivery routes are not suitable for several tumors, including metastatic cancers. In vivo studies with animal models found no correlation between pre-existing immunity and the therapeutic potential of oHSV [99,100]. For efficient systemic delivery of oHSV, the host is usually administered with immunosuppression agents [101]. Although administration of immunosuppression agents such as corticosteroid had no effect on the oncolytic viral activity, they inhibited the establishment of antitumor immunity [102]. Another important issue is the need for combined use of oHSV chemotherapy and/or radiotherapy. These two types of traditional therapy usually cause tissue necrosis and/or induce an inflammatory response that can limit virus spread [103].

Other important issues that should be considered while evaluating the safety profile of oHSV include: (i) the risk of reversion, (ii) latency, and (iii) modulation of the host immune response. Genetically modified variants of oHSV can restore their neurovirulence activity by recombination with WT HSV-1 strain in vivo [103]. The risk of acquiring neurovirulence will lead to a loss of specificity and the infection of healthy tissues. Latent infection is a feature for herpesviruses, and hence oHSV enter latency and reappear later on [104]. oHSVs usually carry one or more of the transgenes that act to modulate the host immune response. Transgenes are added to express several cytokines such as GM–CSF and IL-12, which may cause an inflammatory response against self-antigens and hence autoimmunity [103].

## 7. Conclusions and Perspectives

There has been a marked increase in oHSV research in the last two decades, as genetically modified HSV was the first oncolytic virus to be tested [19]. HSV, in particular, is an attractive agent for oncolytic virotherapy, as its genome is susceptible to modification, providing a promising platform to generate a safe and potent antitumor drug. The success or failure of oncolytic virotherapy is largely dependent on the interaction of antitumor and antiviral immune responses between host and virus [60]. In addition to lysing the tumor cells, oHSVs induce cytokine production and recruitment of immune cells to the tumor microenvironment and enhance antitumor immunity [20]. oHSV has shown efficacy in treating human tumors, and more clinical data support the beneficial role of oHSV therapy in combination with other cancer therapeutics, especially the immunotherapeutic agents [28].

Understanding the interactions between host immunity, the oncolytic viral infection and the tumor are essential for developing better strategies to combat cancer [34]. As oncolytic viral infections activate innate and adaptive immunity, the therapeutic efficacy largely depends on the balance between antiviral immunity, where the immune system clears out the viral infection, and antitumor immunity, where the immune system eliminates tumor cells [28]. The immune responses accompanying oncolytic virotherapy and unique to this system, and special consideration must be taken [34]. Early clearance of oHSV may be circumvented by taking the route and mode of delivery into consideration [61].

Effective delivery of such oncolytic viral agents to tumor sites remains a challenge [105]. Identifying the best route of delivery and vehicle is crucial to optimally engage the patient immune response to mediate effective antitumor immunity [106]. Improving engineered oncolytic viral constructs and testing combined immunotherapeutic agents, coupled with the “Trojan horse” concept, offer various therapeutic possibilities [71]. With more preclinical research going into clinical application, researchers are more likely to achieve more success in understanding the best combination of oHSV variants and other immunetherapeutics, as well as the most suitable route and vehicle for delivery that would ultimately help in winning the fight against cancer.

## Figures and Tables

**Figure 1 cells-10-01541-f001:**
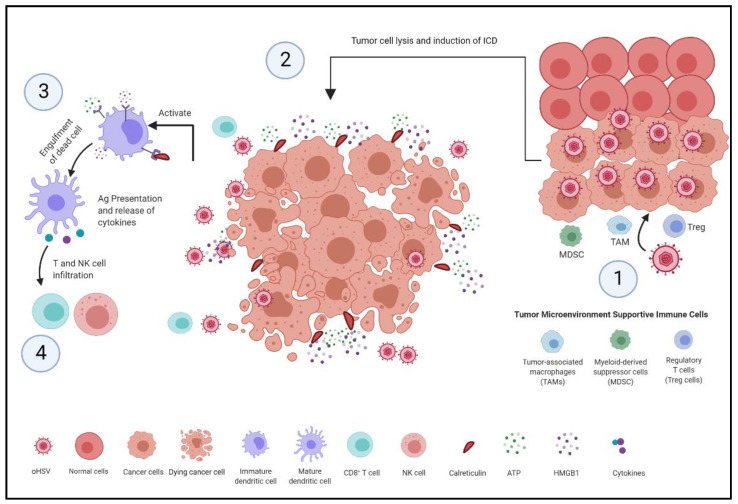
Mechanism of inducing antitumor immunity. (1) oHSV selectively infects and replicates in tumor cells. (2) Infected tumor cells undergo cell lysis and immunogenic cell death (ICD) [28]. The ICD process involves alterations in composition of the cell surface, such as expression of calreticulin (CRT), and the sequential release of damage-associated molecular patterns (DAMPs) such as ATP and high mobility group protein B1 (HMGB1) [20,28]. (3) These alterations and released mediators recruit and interact with receptors expressed on DCs, which recruits them to the tumor site to engulf, process and present tumor antigens [20]. (4) DCs secrete pro-inflammatory cytokines that stimulate natural killer cells and tumor-specific CD8^+^ T cells, thereby inducing immune-mediated tumor cell killing [20].

**Figure 2 cells-10-01541-f002:**
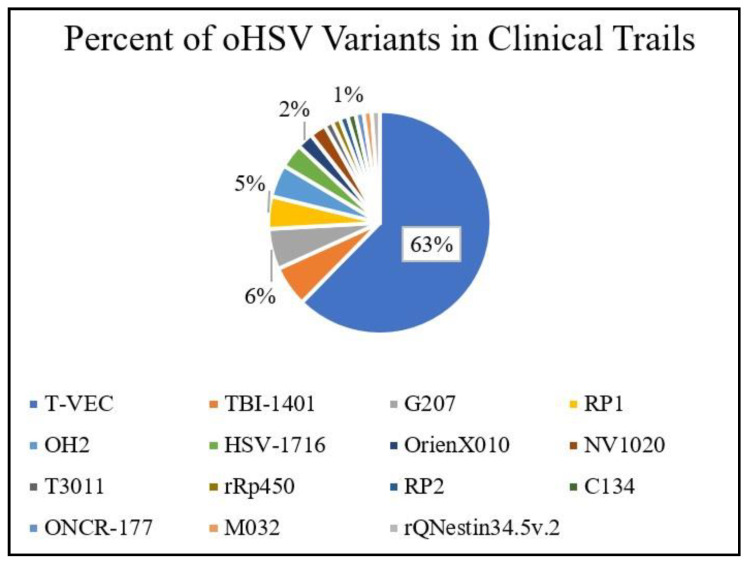
A summary of clinical trials registered at ClinicalTrials.gov involving oHSV variants. Access date: 13 February 2021.

**Table 1 cells-10-01541-t001:** A summary of oHSV variants and their applications in preclinical models.

oHSV Variant	Genetic Changes	Preclinical Models
rRp450	ICP6 deletion, insertion of CYP2B1.	Glioblasoma [42] Colon carcinoma [43] Sarcoma [44]
rQNestin34.5v.2	Nestin promoter drives the expression of ICP34.5.	Glioma [45]
G207	Diploid deletion of γ34.5, inactivation of UL39.	Glioma [46,47] Cervical cancer [48]
G47Δ	Diploid deletion of γ34.5, inactivation of UL39, deletion of α47.	Meningioma [49] Gastic cancer [50] Liver cancer [51] Breast cancer [52] Thyroid carcinoma [53] Urological cancers [54]
G47Δ-TRAIL	G47Δ expressing TRAIL	Glioblastoma [41]
oHSV-IL12	G207 expressing IL12	Ovarian cancer [55] Breast cancer [56]
oHSV-GM-CSF	G207 expressing GM-CSF	Advanced melanoma [18]

**Table 2 cells-10-01541-t002:** A summary of clinical trials registered at ClinicalTrials.gov involving oHSV variants. Access date: 13 February 2021.

Variant	Combination Therapy	Clinical Trial Number	Status	Virus	Malignant Type	Phase	Year Posted
T-VEC	-	NCT00289016	Completed	HSV-1	Stage IIIC and IV melanoma	II	2006
	-	NCT00289016	Completed	HSV-1	Stage IIIC and IV melanoma	II	2006
	-	NCT00769704	Completed [89]	HSV-1	Melanoma	III	2008
	Cisplatin + Radiation	NCT01161498	Terminated	HSV-1	Head and neck cancer	III	2010
	GM-CSF	NCT01368276	Completed	HSV-1	Melanoma	III	2011
	Ipilimumab	NCT01740297	Active, not recruiting [90]	HSV-1	Melanoma	I, II	2012
	-	NCT02014441	Completed [87]	HSV-1	Melanoma	II	2013
	Pembrolizumab	NCT02263508	Active, not recruiting	HSV-1	Melanoma	III	2014
	-	NCT02173171	Enrolling by invitation	HSV-1	Any tumor type	Unspecified	2014
	Resection surgery	NCT02211131	Active, not recruiting	HSV-1	Melanoma	II	2014
	Pembrolizumab	NCT02263508	Active, not recruiting	HSV-1	Melanoma	III	2014
	-	NCT02297529	No longer available	HSV-1	Stage IIIB-IVM1c melanoma	III	2014
	Chemotherapy or PV-10	NCT02288897	Terminated	HSV-1	Melanoma	III	2014
	-	NCT02366195	Completed	HSV-1	Stage IIIb-IVM1c melanoma	II	2015
	Radiation	NCT02453191	Active, not recruiting	HSV-1	Soft tissue sarcoma	I, II	2015
	Pembrolizumab	NCT02626000	Completed [91]	HSV-1	Squamous cell carcinoma of the head and neck	I	2015
	-	NCT02574260	Completed	HSV-1	Melanoma	II	2015
	Pembrolizumab	NCT02509507	Recruiting	HSV-1	Liver cancer	I	2015
	-	NCT02658812	Active, not recruiting	HSV-1	Breast cancer	II	2016
	-	NCT02756845	Recruiting	HSV-1	Advanced non-CNS tumors	I	2016
	Radiation	NCT02819843	Recruiting	HSV-1	Melanoma, Markel cell carcinoma, and other tumors	II	2016
	Radiation	NCT02923778	Recruiting	HSV-1	Soft tissue sarcoma	II	2016
	Nivolumab	NCT02978625	Recruiting	HSV-1	Melanoma, lymphoma, lung cancer, and other	II	2016
	Pembrolizumab	NCT02965716	Recruiting	HSV-1	Stage III and IV melanoma	II	2016
	Paclitaxel	NCT02779855	Active, not recruiting	HSV-1	Triple negative breast cancer	I, II	2016
	-	NCT02658812	Completed	HSV-1	Breast cancer	II	2016
	-	NCT02910557	Recruiting	HSV-1	Melanoma and herpetic infection	Unspecified	2016
	TTI-621	NCT02890368	Terminated	HSV-1	Solid tumors and mycosis fungoides	I	2016
	-	NCT03086642	Recruiting	HSV-1	Pancreatic cancer	I	2017
	-	NCT03064763	Active, not recruiting	HSV-1	Stage IIIb- IV melanoma	I	2017
	Pembrolizumab	NCT03069378	Recruiting	HSV-1	Sarcoma	II	2017
	Atezolizumab	NCT03256344	Active, not recruiting	HSV-1	Metastatic colorectal and breast cancers	I	2017
	Chemotherapy + Radiation	NCT03300544	Recruiting	HSV-1	Rectal cancer	I	2017
	Dabrafenib + Trametinib	NCT03088176	Active, not recruiting	HSV-1	Melanoma	I	2017
	-	NCT03458117	Unknown	HSV-1	Non-melanoma skin cancer	I	2018
	-	NCT03555032	Active, not recruiting	HSV-1	Sarcoma and melanoma	I, II	2018
	-	NCT03663712	Recruiting	HSV-1	Stage IV peritoneal malignancy	I	2018
	Autologous CD1c (BDCA-1)^+^ myeloid dendritic cells	NCT03747744	Active, not Recruiting	HSV-1	Melanoma	I	2018
	Nivolumab	NCT03597009	Recruiting	HSV-1	Lung cancer	I, II	2018
	-	NCT03430687	Withdrawn	HSV-1	Bladder carcinoma	I	2018
	-	NCT03714828	Recruiting	HSV-1	Squamous cell carcinoma	II	2018
	Chemotherapy or endocrine therapy	NCT03554044	Recruiting	HSV-1	Breast cancer	I	2018
	-	NCT03921073	Recruiting	HSV-1	Skin angiosarcoma	II	2019
	Atezolizumab	NCT03802604	Recruiting	HSV-1	Breast cancer	I	2019
	Pembrolizumab	NCT03842943	Recruiting	HSV-1	Melanoma	II	2019
	Dabrafenib + Trametinib	NCT03972046	Withdrawn	HSV-1	Melanoma	II	2019
	Pembrolizumab	NCT04068181	Recruiting	HSV-1	Melanoma	II	2019
	Panitumumab	NCT04163952	Recruiting	HSV-1	Squamous cell carcinoma of the skin	I	2019
	Nivolumab + Trabectedin	NCT03886311	Recruiting	HSV-1	Sarcoma	II	2019
	-	NCT04065152	Not yet recruiting	HSV-1	Kaposi sarcoma	II	2019
	Ipilimumab + Nivolumab	NCT04185311	Recruiting	HSV-1	Breast cancer	I	2019
	-	NCT04330430	Recruiting	HSV-1	Stage III and IV melanoma	II	2020
TBI-1401	-	NCT01017185	Completed	HSV-1	Squamous cell carcinoma of the skin, breast carcinoma, melanoma, head and neck cancer	I	2009
	Ipilimumab	NCT02272855	Completed	HSV-1	Melanoma	II	2014
	-	NCT02428036	Completed	HSV-1	Solid tumors	I	2015
	Ipilimumab	NCT03153085	Completed	HSV-1	Stage III and IV melanoma	II	2017
	Chemotherapy	NCT03252808	Active, not recruiting	HSV-1	Stage III and IV pancreatic cancer	I	2017
G207	-	NCT00028158	Completed	HSV-1	Brain cancer	I, II	2001
	Radiation	NCT00157703	Completed	HSV-1	Malignant glioma	I	2005
	-	NCT02457845	Active, not recruiting	HSV-1	Brain cancer	I	2015
	-	NCT03911388	Recruiting	HSV-1	Brain cancers	I	2019
	-	NCT04482933	Not yet recruiting	HSV-1	High grade glioma	II	2020
RP1	Nivolumab	NCT03767348	Recruiting	HSV-1	Melanoma	I, II	2018
	Cemiplimab	NCT04050436	Recruiting	HSV-1	Melanoma	II	2019
	-	NCT04349436	Recruiting	HSV-1	Squamous cell carcinoma	I	2020
RP2	Nivolumab	NCT04336241	Recruiting	HSV-1	Non-specified	I	2020
HSV-1716	-	NCT00931931	Completed [92]	HSV-1	Non-CNS solid tumors	I	2009
	-	NCT01721018	Completed	HSV-1	Malignant pleural mesothelioma	I, II	2012
NV1020	-	NCT00012155	Completed	HSV-1	Colorectal cancer	I	2003
	-	NCT00149396	Completed [93]	HSV-1	Liver cancer and colorectal cancer	I, II	2005
OrienX010	-	NCT01935453	Completed	HSV-1	Melanoma, liver cancer, pancreatic cancer and lung cancer	I	2013
	-	NCT03048253	Unknown	HSV-1	Melanoma	I-c	2017
rRp450	-	NCT01071941	Recruiting	HSv-1	Liver cancer	I	2010
M032	-	NCT02062827	Recruiting	HSV-1	Brain cancers	I	2014
rQNestin	-	NCT03152318	Recruiting	HSV-1	Brain cancers	I	2017
C134	-	NCT03657576	Active, not recruiting	HSV-1	Malignant glioma	I	2018
	T3011	NCT04370587	Recruiting	HSV-1	Head and neck cancer, Melanoma, Lung cancer, Soft tissue tumors and/or sarcoma, Solid tumors	I	2020
ONCR-177	Pembrolizumab	NCT04348916	Recruiting	HSV-1	Various tumors	I	2020
OH2	HX008	NCT03866525	Recruiting	HSV-2	Solid and GI tumors	I, II	2019
	Pembrolizumab	NCT04386967	Recruiting	HSV-2	Solid tumors	I, II	2020
	-	NCT04637698	Recruiting	HSV-2	Pancreatic cancer	I, II	2020
	HX008	NCT04616443	Recruiting	HSV-2	Melanoma	I, II	2020

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
