# Peer review of "Oncolytic Herpes Simplex Virus-Based Therapies for Cancer"

_cells, 2021, doi:10.3390/cells10061541_

Round 1

Reviewer 1 Report

While oncolytic viruses have been under intense development over the last decade, an oncolytic HSV vector, T-Vec (Imlygic), is currently the only oncolytic virus that has gained market approval as a cancer therapeutic in the U.S. and Europe. As a result of this success, numerous other HSV-based vectors have been engineered and tested as part of novel cancer immunotherapeutic combinatorial approaches. This review discusses the mechanism of action of these viruses, summarizes the further development of HSV vectors, and provides an overview of the clinical trials involving HSV vectors. In general, it is an informative review article for anyone looking for an overview of HSV as an oncolytic virus.

I have just a few comments:

  • The manuscript would be more interesting to read if the approval of T-Vec was mentioned earlier. The authors should also go into some detail about the effectiveness (and short-comings) of T-Vec.
  • As there are numerous other OV platforms under development, a stronger justification for the use of HSV should be provided in the introduction. What are the benefits of HSV compared to other virus platforms?
  • The description of innate immune responses and the mechanism of anti-tumor immune responses (including Figure 1) are not specific for HSV and are generally described as the mechanism of action for all OVs. It would be helpful if some specific examples or data on HSV were provided here.
  • There are currently numerous clinical trials investigating the use of T-Vec with immune checkpoint blockade therapies. A short discussion of these trials and a summary of any available clinical findings of these studies would be relevant, as well as a review of the numerous preclinical studies combining HSV with ICB as a combination therapy.

Reviewer 2 Report

The review contains up-to-date information on approaches to the creation of anticancer drugs based on modified herpes simplex viruses. Oncolytic viruses are a relatively new tool in the treatment of cancer. A real breakthrough in this direction was achieved in 2015, when the first oncolytic virus was approved by the FDA for the treatment of unresectable stage III advanced melanoma. This first oncolytic virus is T-VEC, a modified herpes simplex virus type 1. Thus, it is thanks to the herpes simplex virus that oncolytic viruses have gained recognition in the human community.

The text of the review requires a minor revision, starting from the title. The emphasis on HSV-1 only should be removed, since the review contains information on oncolytic viruses based on HSV types 1 and 2. The name might sound like this: Oncolytic Herpes Simplex Virus-based Therapies for Cancer.

Section 2.1. Tumor-selective Replication is not very clear and too short. In particular, it is not clear what mechanisms provide tumor selectivity in the case of deletion of the γ34.5 and α47 genes.

The material in Table 2 should be somehow systematized, for example, to arrange the clinical trials by increasing the year posted, or by combining them into subsections for each oHSV variant.

Table 2 shows very interesting data from the latest (2019-2020) clinical trials of modified HSV-2, but these drugs are not described at all in the review text. It is desirable to describe briefly what they are.

Care should be taken throughout the text to check the spelling of words (Fig.2 – varients) and references. For example, a reference on page 4, line 163 (Yin et al., 2017) should be replaced by [17] and a reference on line 164 (Gujar & Lee, 2014) should be replaced by [28].
